# Low yield and abiotic origin of $N_2O$ formed by the complete nitrifier *Nitrospira inopinata*

K. Dimitri Kits[1], Man-Young Jung[1], Julia Vierheilig [1,6,7], Petra Pjevac[1], Christopher J. Sedlacek[1], Shurong Liu[1,2], Craig Herbold[1], Lisa Y. Stein[3], Andreas Richter [2,4], Holger Wissel[5], Nicolas Brüggemann[5], Michael Wagner [1,2] & Holger Daims[1,2]

Nitrous oxide ($N_2O$) and nitric oxide (NO) are atmospheric trace gases that contribute to climate change and affect stratospheric and ground-level ozone concentrations. Ammonia oxidizing bacteria (AOB) and archaea (AOA) are key players in the nitrogen cycle and major producers of $N_2O$ and NO globally. However, nothing is known about $N_2O$ and NO production by the recently discovered and widely distributed complete ammonia oxidizers (comammox). Here, we show that the comammox bacterium *Nitrospira inopinata* is sensitive to inhibition by an NO scavenger, cannot denitrify to $N_2O$, and emits $N_2O$ at levels that are comparable to AOA but much lower than AOB. Furthermore, we demonstrate that $N_2O$ formed by *N. inopinata* formed under varying oxygen regimes originates from abiotic conversion of hydroxylamine. Our findings indicate that comammox microbes may produce less $N_2O$ during nitrification than AOB.

[1] Centre for Microbiology and Environmental Systems Science, Division of Microbial Ecology, University of Vienna, Althanstrasse 14, 1090 Vienna, Austria. [2] The Comammox Research Platform, University of Vienna, Althanstrasse 14, 1090 Vienna, Austria. [3] Department of Biological Sciences, University of Alberta, CW405 Biological Sciences Building, Edmonton, AB T6G 2E9, Canada. [4] Centre for Microbiology and Environmental Systems Science, Division of Terrestrial Ecosystem Research, University of Vienna, Althanstrasse 14, 1090 Vienna, Austria. [5] Institute of Bio- and Geosciences-Agrosphere (IBG-3), Forschungszentrum Jülich GmbH, 52425 Jülich, Germany. [6] Present address: Karl Landsteiner University of Health Sciences, Division of Water Quality and Health, Krems 3500, Austria. [7] Present address: Interuniversity Cooperation Centre for Water and Health, Krems 3500, Austria. Correspondence and requests for materials should be addressed to M.W. (email: wagner@microbial-ecology.net)

Nitrous oxide ($N_2O$) is the third most abundant greenhouse gas in the atmosphere. It contributes ~6% to the total radiative forcing and is also predicted as the dominant ozone depleting substance throughout the 21st century[1]. The atmospheric $N_2O$ concentration has continuously increased over the last decades at an average rate of ~0.31% per year, and this trend will continue[1,2]. Anthropogenic emissions of $N_2O$ make up 30–45% of the total global budget, with about two-thirds of this coming from agricultural and soil sources[1], and are predicted to increase in the future as application of nitrogen fertilizers rises to feed the growing human population[1,3]. Microbial transformations of nitrogenous compounds, especially heterotrophic denitrification and chemolithoautotrophic aerobic nitrification, are the dominant contributor to $N_2O$ emissions from agriculture and soil management as well as wastewater treatment, the latter of which adds ~3.4% to the global $N_2O$ emission budget[1,3]. In addition to $N_2O$, denitrifying and nitrifying microbes also release nitric oxide (NO) that represents an important metabolic intermediate for both guilds[4]. This activity is also environmentally relevant as NO contributes to the production of ground-level ozone and acid rain[5]. Thus, understanding the microbial players involved in NO/$N_2O$ production, the pathways that lead to the generation of these gases, and the environmental factors that control their fluxes is critical to modeling future emissions and developing appropriate mitigation strategies.

Classically, nitrification has been thought of as a two-step process. Ammonia ($NH_3$) is oxidized via hydroxylamine ($NH_2OH$) to nitrite ($NO_2^-$) by AOB and AOA, and subsequently nitrite oxidation to nitrate ($NO_3^-$) is catalyzed by nitrite oxidizing bacteria (NOB)[4]. Within the ammonia oxidizing microbes, two pathways were traditionally thought to contribute to NO and $N_2O$ emissions: (1) aerobic $N_2O$ formation from the abiotic reaction of the intermediate $NH_2OH$ with $NO_2^-$ (also referred to as hybrid $N_2O$ formation), and (2) enzymatically catalyzed $NO_2^-$ reduction to $N_2O$ via NO through "nitrifier-denitrification"[6,7]. The first pathway has been described for AOB and AOA[8,9], while the latter has only been reported for AOB[8,10]. In AOB, hybrid $N_2O$ formation is the dominant process at atmospheric oxygen levels, while nitrifier-denitrification is more important at low $O_2$ tension[9,11–13]. Very recently, two additional routes for NO/$N_2O$ production from AOB have been characterized. Firstly, the periplasmic tetraheme cyt. $c$ P460 protein (CytL, present in most but not all AOB) oxidizes two molecules of $NH_2OH$ to $N_2O$ and water under anoxic conditions[14], although the kinetics of this protein may render it inefficient at this role under physiological conditions. Additionally, this protein can bind NO and reduce it

in the presence of $NH_2OH$ to $N_2O$[14]. Second, NO (and not $NO_2^-$ as previously considered) is formed by the activity of hydroxylamine dehydrogenase (HAO) under oxic and anoxic conditions[14,15]. CytL is used by AOB to detoxify $NH_2OH$ and NO, while the activity of HAO leading to NO formation, which is further oxidized by an unknown enzyme to $NO_2^-$, is essential for energy conservation in these organisms (Fig. 1). Pure culture work on marine and soil AOA, and soil microcosm studies on complex communities, strongly suggest that archaea produce lower yields of $N_2O$ than AOB ($N_2O/NO_2^-$ ratio %; 0.04–0.07% for AOA, 0.095–0.27% for AOB) during aerobic ammonia oxidation[8,16–18].

Two-step nitrification requires coupling between ammonia oxidation and nitrite oxidation. Consequently, the two steps can also become uncoupled, for example, under high nitrogen load, which can inhibit NOB[19,20]. Additionally, mismatched nutrient affinities for $NH_3$ and $NO_2^-$ between the two guilds and varying energetic constraints often allow for the accumulation of $NO_2^-$[20–22]. The accumulated $NO_2^-$ from uncoupled nitrification can drive production of $N_2O$ from hybrid formation, nitrifier-denitrification, and heterotrophic denitrification. In addition, it is important to keep in mind that NO and $N_2O$ can also be produced by a multitude of chemical reactions that use the key metabolites of ammonia oxidizers – $NH_2OH$ and $NO_2^-$ (or its protonated form $HNO_2$) – as the main precursors (for recent reviews see ref. [5]).

Recently, the traditional perspective that the two steps of nitrification are always catalyzed by different microorganisms was refuted by the discovery of 'comammox' organisms that oxidize ammonia to nitrate on their own[23,24]. So far, comammox is restricted to the phylogenetic lineage II within the bacterial genus *Nitrospira*, which also contains canonical NOB[25]. Comammox genomes contain all genes encoding the known bacterial machineries for ammonia and nitrite oxidation – ammonia monooxygenase (AMO), HAO, the hydroxylamine ubiquinone reduction module (HURM), and nitrite oxidoreductase (NXR)[23,24,26,27]. Comammox *Nitrospira* are widely distributed and have been detected in various environments including pristine and agricultural soils, freshwater habitats, drinking water treatment systems, aquaculture biofilters, hot groundwater, and wastewater treatment plants in which they thrive, sometimes at considerable abundance[23,24,27–31].

Interestingly, a potential solution to the aforementioned synchrony problem between canonical AOB/AOA and NOB is naturally provided by the comammox organisms which perform ammonia oxidation and nitrite oxidation within one cell. Consistently, recent mathematical modeling has suggested that comammox organisms should show improved efficiency in nitrogen removal and a significantly reduced production of NO and $N_2O$[32]. However, it is currently not known whether comammox bacteria actually produce NO and/or $N_2O$ and if yes at which yields and via which pathways.

Recently, we obtained a pure culture of a comammox organism (*Nitrospira inopinata*) and characterized it kinetically[33]. In this study, we use this pure culture to: (1) quantify NO and $N_2O$ production by *N. inopinata*, (2) determine which biotic/abiotic pathways contribute to potential NO/$N_2O$ production in this organism, (3) compare the yields of NO/$N_2O$ from *N. inopinata* with known values for AOA and AOB, and (4) compare the genomic inventory for reactive nitrogen metabolism in various comammox organisms to gain insight into potential heterogeneity of their NO and $N_2O$ production pathways. By using a combination of micro-respirometry, gas chromatography, NO scavenging assays, $N_2O$ isotope analysis, and comparative genomics we demonstrate that *Nitrospira inopinata*, and possibly all other currently (meta)genomically characterized comammox

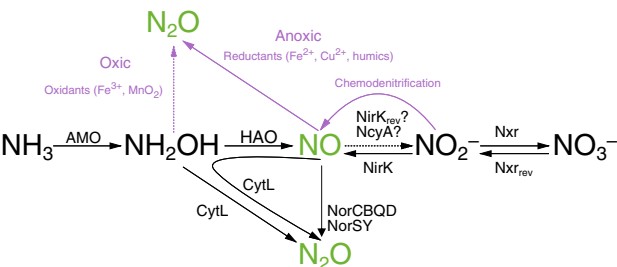

**Fig. 1** Biotic and abiotic pathways leading to NO and $N_2O$ production in bacterial nitrifiers. Black arrows depict confirmed (solid) and proposed (dashed) enzymatic reactions; confirmed or proposed enzymes are noted above or below the arrow. NcyA and NirK$_{rev}$ are candidates for the NO-oxidizing (NOO) enzyme bacterial ammonia-oxidizers. Violet arrows represent abiotic reactions that occur under oxic (dashed) or anoxic (solid) conditions with violet text outlining the reactants/conditions that favor those processes

organisms, produce NO as an important intermediate but cannot denitrify to $N_2O$ and thus produce low yields of $N_2O$ that are comparable to soil AOA.

## Results and discussion

**Comammox *Nitrospira* lack NO reductases**. NO and $N_2O$ production by AOB, AOA, and NOB has been intensively studied and several key genes used by these nitrifiers for the production and consumption of these gases have been identified[8,11,34–38] (Figs. 1 and 2). In contrast, no experimental data about the ability of comammox organisms to form NO and $N_2O$ are available and nothing is known about the potential importance of these compounds for the metabolism of complete nitrifiers.

As a first step to close this knowledge gap, we updated previous comparative genomic analyses of comammox genomes[26,27] and mined them for key genes that might be involved in NO and $N_2O$ metabolism (Fig. 2). These analyses included *N. inopinata*, the only comammox organism available as pure culture[33], as well as genomes from two comammox enrichments and 15 comammox metagenome-assembled genomes (MAGs) from environmental samples retrieved via metagenomics[23,26,29,30,39]. All comammox organisms encode ammonia oxidation machinery that is more closely related to AOB than to AOA[24]. Part of this machinery is HAO that, in AOB, converts hydroxylamine (formed from $NH_3$ by the AMO in an oxygen-dependent manner) to NO via an $O_2^-$ independent, three-electron oxidation step[15,24]. Given its phylogenetic relationship to the HAO of AOB, and the fact that a planctomycete enzyme that shares multiple characteristics with HAO from *N. europaea* also produces NO[40], it seems highly likely that the HAO of comammox organisms also generates NO. Under this assumption, comammox organisms, like AOB, would benefit from the enzymatic oxidation of NO to $NO_2^-$ via an unknown NO oxidoreductase (NOO)[41] in order to harvest a fourth electron for electron transport. For AOB the most compelling candidate for NOO is the red copper protein nitrosocyanin (NcyA), which is present in most AOB and is as highly expressed as AMO and HAO[41,42]. However, none of the comammox *Nitrospira* genomes we analyzed encoded *ncyA*, suggesting an alternative candidate for NOO (Fig. 2). All of the genomes, with the exception of a single MAG (with an estimated completeness of 86.84%), contained the *nirK* gene. NirK is encoded by many (but not all) AOB, AOA, and NOB[6,27,43–46] and catalyzes the one electron reduction of $NO_2^-$ to NO. NO generation by this enzyme is used by some AOB to facilitate efficient $NH_3$ oxidation and also under hypoxic conditions to enable intracellular redox-balance via nitrifier-denitrification (Supplementary Note 1)[11,13,37,47]. Although NirK has been shown to operate reversibly and oxidize NO to $NO_2^-$[48], the kinetics of the reaction are highly unfavorable at intracellular pH and redox potential, arguing that NirK is not an ideal candidate for the NOO.

In many AOB, the two-electron reduction of two molecules of NO to $N_2O$ is performed by two classes of cytochrome *c* nitric oxide reductase (NOR) – *norCBQD* and the alternative NO reductase *norSY* – while all genome-sequenced AOA lack NOR[10,27]. Physiological analyses of several AOB strains with or without cytochrome *c*-dependent NORs demonstrated that these enzymes are required for $N_2O$ formation via nitrifier-denitrification[6,37]. Interestingly, homologs for *norCBQD* and *norSY* are absent from all 23 currently known comammox MAGs and also absent from genomes of all cultivated strains including *N. inopinata* (Fig. 2)[27,33]. Comammox bacteria might thus not be able to reduce NO to $N_2O$ as part of the nitrifier-denitrification pathway under hypoxic conditions, similar to oligotrophic strains of AOB[6].

We also queried the presence of the *c* cytochromes P460 (*cytL*) and *c′*-beta (*cytS*) in the genomes of comammox *Nitrospira*. The P460 enzyme from *N. europaea* is considered to be a detoxifying enzyme[14] as it converts two equivalents of $NH_2OH$ (or one $NH_2OH$ and one NO) to $N_2O$ under anoxic conditions, though this enzyme is expressed during aerobic growth in some but not all tested AOB[14,42]. The function of CytS is still unknown, but for AOB it has been suggested to be involved in the oxidation/reduction of N-oxides or electron transfer for either detoxification or energy conservation[49]. Both the cytochrome P460 and cytochrome *c′*-beta are found sporadically in genomes of comammox *Nitrospira* (Fig. 2). All of the currently enriched or cultured comammox representatives including *N. inopinata* lack *cytL*, while the genomes of *N. inopinata* and *Ca.* N. nitrosa contain the uncharacterized *cytS* (Fig. 2).

Taken together, *N. inopinata* and all other genome-sequenced comammox microbes possess the genetic potential to produce NO via HAO or NirK activity, but lack *bona fide* NO reductases to form $N_2O$. However, keeping in mind that ~46% of the *N. inopinata* genes have no functional annotation and several, often unrelated, enzyme classes can catalyze identical transformations of nitrogen compounds[50], physiological experiments as well as protein purification and characterization are clearly required to examine formation, magnitude, and importance of NO and $N_2O$ formation and NO oxidation in comammox organisms.

***N. inopinata* releases and consumes NO under oxic conditions**. To determine whether *N. inopinata* produces and consumes NO, instantaneous $O_2$ and NO kinetics were measured with micro-sensors during $NH_3$ and $NO_2^-$ oxidation by *N. inopinata*. Addition of 250 μM $NH_4^+$ into the micro-respirometry (MR) chamber led to immediate substrate-dependent $O_2$ consumption and NO production. NO production peaked at ~13 nM after ~30% of the dissolved $O_2$ was consumed, followed by net NO consumption (Fig. 3). Since *N. inopinata* is an oligotroph, we also tested how substrate concentration influences net NO flux. Lower substrate concentrations (<15 μM $NH_4^+$) resulted in significantly less NO production (<0.8 nM) (Supplementary Fig. 1). There was no measurable NO production after $O_2$ was depleted in the presence of $NH_3$, reflecting the dependency of NO production on $O_2$ and strongly suggesting that *N. inopinata* under the conditions applied did not respire anaerobically with $NO_2^-$ as an electron acceptor using storage products as electron donors. Despite comparable absolute cell numbers (~$1 \times 10^{10}$ cells), net production of NO (30 nM per ~$1 \times 10^{10}$ cells) by the *N. inopinata* biomass was about an order of magnitude lower during aerobic oxidation of the same amount of $NH_3$ than by the oligotrophic AOB *Nitrosomonas* sp. Is79A3 or *N. ureae*[6]. Furthermore, previous work demonstrated that both of these oligotrophic AOB produce very large (>250 nM) quantities of NO after the onset of hypoxia (ascribed to the activity of their NirK enzymes). This phenomenon reflects the absence of NO reductases in these strains, which are required for NO reduction to $N_2O$ in the nitrifier-denitrification pathway[6,37]. In contrast, the NO reductase encoding *Nitrosomonas europaea* ATCC 19718 produced ~180–210 nM NO (per ~$1 \times 10^{10}$ cells) throughout aerobic $NH_3$ oxidation and consumed NO via nitrifier denitrification after the onset of hypoxia (Supplementary Fig. 2). Consistently, other previously analyzed AOB also produced >50 nM NO (per $1 \times 10^{10}$ total cells) during aerobic $NH_3$ oxidation prior to hypoxia in the MR chamber[6].

The lack of measurable NO production by *N. inopinata* during $O_2$-limited conditions despite the fact that NirK was the second most abundant protein in proteomic analysis of *N. inopinata* even under aerobic conditions (Supplementary Fig. 3) suggests that

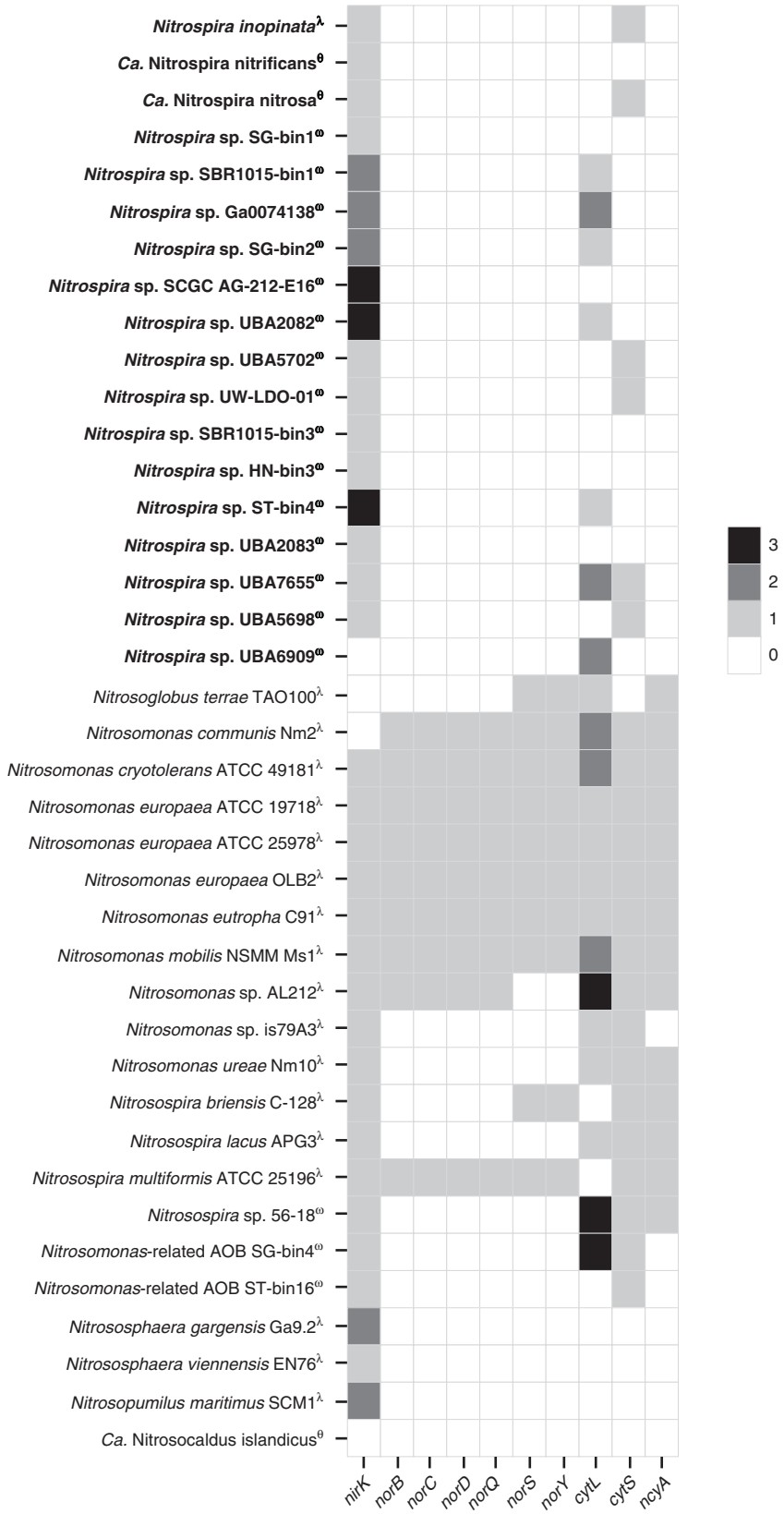

**Fig. 2** Gene inventory implicated in $NO_x$ metabolism in publicly available genomes of various nitrifiers. Comammox *Nitrospira* genomes are depicted in boldface. Genomes from enrichments and pure cultures are denoted by the symbols θ and λ, respectively. The symbol ω denotes uncultured organisms. The number of copies of each gene are denoted by the color bar. Published genomes and MAGs were compiled from the literature (see Supplementary Data 1 for sources, source data, locus tags, and bin/genome statistics) and downloaded from the Integrated Microbial Genomes website (https://img.jgi. doe.gov/cgi-bin/mer/main.cgi)

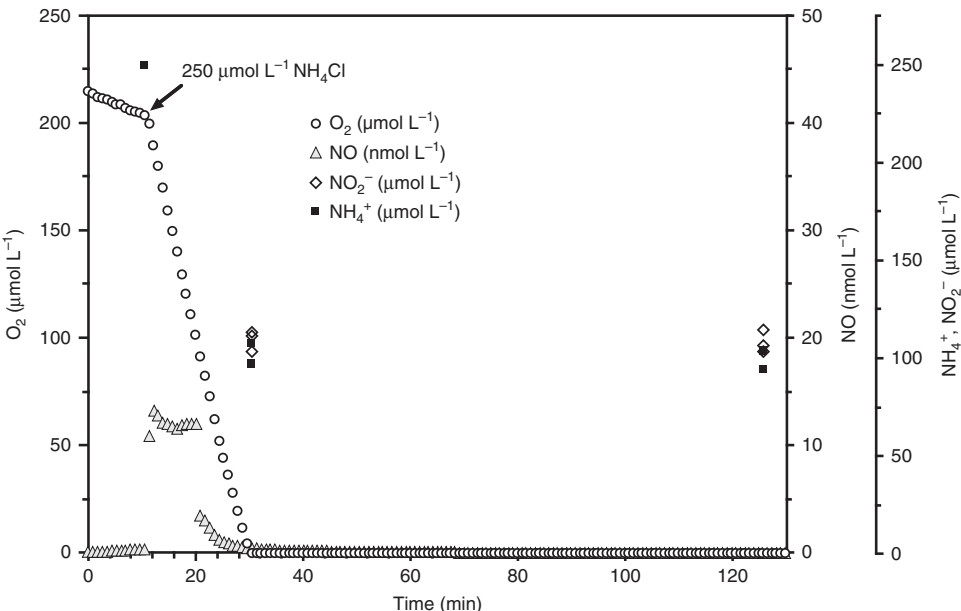

**Fig. 3** Instantaneous $O_2$ consumption and NO production during $NH_3$ oxidation by *N. inopinata*. The data shown here is a single representative of three biological replicates ($n = 3$). Two additional biological replicates are shown in Supplementary Fig. 6. Dissolved $O_2$ is shown in open circles, dissolved NO in filled gray triangles, $NO_2^-$ in open diamonds, and $NH_4^+$ in filled squares. The $NH_4^+$ concentration immediately after injection (~10 min) was inferred from the injected volume of a stock $NH_4Cl$ solution, otherwise $NO_2^-$ and $NH_4^+$ concentrations were determined in three technical replicates ($n = 3$). Experiments were performed in a microrespiration (MR) chamber fitted with $O_2$ and NO microsensors. The arrow marks the addition of 250 μM $NH_4Cl$ into the MR chamber. About 110 μM residual $NO_2^-$ was present in the chamber once $O_2$ reached a concentration below the detectable level at ~35 min. No NO formation from $NH_4^+$ was measurable in sterile media controls containing the same amount of heat-killed biomass of *N. inopinata*. Source data are provided as a Source Data file

NirK either has very weak activity under hypoxic conditions, producing NO below the detection limit of our instrument (~0.25 nM NO), or that *N. inopinata* does not perform NirK-based nitrifier-denitrification during hypoxia at all. The NO profile during aerobic $NH_3$ oxidation by *N. inopinata* shows interesting similarities to that previously reported for the AOA strain *N. viennensis*, which also rapidly produces NO and then consumes it during aerobic $NH_3$ oxidation[10]. However, in contrast to *N. inopinata*, *N. viennensis* additionally produces NO at the onset of hypoxia[10]. Although the NO production and consumption profiles of *N. inopinata* and the AOA strains *N. viennensis* and *N. maritimus* are different[10,51,52], NirK is strongly expressed in all three organisms[53,54] and the release of NO appears to be more tightly controlled by them than in AOB.

Interestingly, using $NO_2^-$ instead of $NH_3$ as the electron donor resulted in a different NO production and consumption profile by *N. inopinata* (Fig. 4); addition of 2.5 mM $NO_2^-$ led to immediate net NO production after which NO levels reached a steady-state level of ~45 nM (Fig. 4). Net consumption of NO was not evident during further $NO_2^-$ oxidation. In contrast, instantaneous NO concentrations were below our detection limit (~0.25 nM) for the closely related non-comammox *Nitrospira*, *N. moscoviensis*, during $NO_2^-$ oxidation (Supplementary Fig. 4). To our knowledge, NO production in *Nitrospira* has not been investigated previously but previous work has shown that NO (65 nM NO in the liquid phase) inhibits $NO_2^-$ oxidation to $NO_3^-$ in *Nitrospira* dominated sludge[55]. The more distantly related proteobacterial nitrite-oxidizer, *Nitrobacter winogradskyi* produces[38] and consumes NO[36,38] in a $NO_2^-$-dependent manner during normal aerobic growth. However, the total NO flux is strongly influenced by its quorum sensing system and decreased when quorum sensing was quenched[38].

It is interesting to note that in *N. winogradskyi* the NO producing nitrite reductase NirK has been postulated to facilitate nitrite oxidation under $O_2$-limiting conditions by maintaining redox balance via regulation of electron flow. At low oxygen tension in the presence of nitrite, NirK is strongly upregulated and more NO is produced causing reversible inhibition of the heme-copper cyt *c* terminal oxidase and thus intensifying reverse electron transport for $NAD(P)^+$ reduction and storage compound formation[36]. In contrast, we show here that stable NO production by *N. inopinata* during $NO_2^-$ oxidation occurs under fully oxic conditions, which is consistent with a constitutive high expression of NirK during aerobic growth of *N. inopinata* (Fig. 4) (Supplementary Fig. 3). Alternatively, it is conceivable that the contrasting NO profiles in *N. moscoviensis* and *N. inopinata* are caused by a reversal of the yet unknown NOO enzyme in *N. inopinata* at very high $NO_2^-$ concentrations, a process that does not occur in *N. moscoviensis* because it lacks the genetic repertoire to oxidize $NH_3$ to $NO_2^-$.

**PTIO is a potent inhibitor of $NH_3$ oxidation in *N. inopinata*.** Both AOB and AOA produce NO as an obligate intermediate during $NH_3$ oxidation[10,41,51,52]. However, AOA exhibit very tight control over the production and consumption of NO during $NH_3$ oxidation[10,51,52]. This difference in the regulation of NO concentrations may explain why the AOA are selectively inhibited by low concentrations of the NO scavenger PTIO[51,56]. The similar aerobic NO kinetics between *N. inopinata* and the AOA *N. viennensis* raised the hypothesis that comammox *Nitrospira* may be sensitive to inhibition by low concentrations of PTIO as well. We tested this hypothesis by comparing instantaneous $O_2$ consumption rates and batch culture $NH_3$ oxidation activity of *N. inopinata* in the presence of various concentrations of PTIO to non PTIO-amended biomass (Supplementary Fig. 5). Indeed, PTIO was a potent inhibitor of instantaneous $O_2$ consumption and batch culture $NH_3$ oxidation in *N. inopinata* with a half-

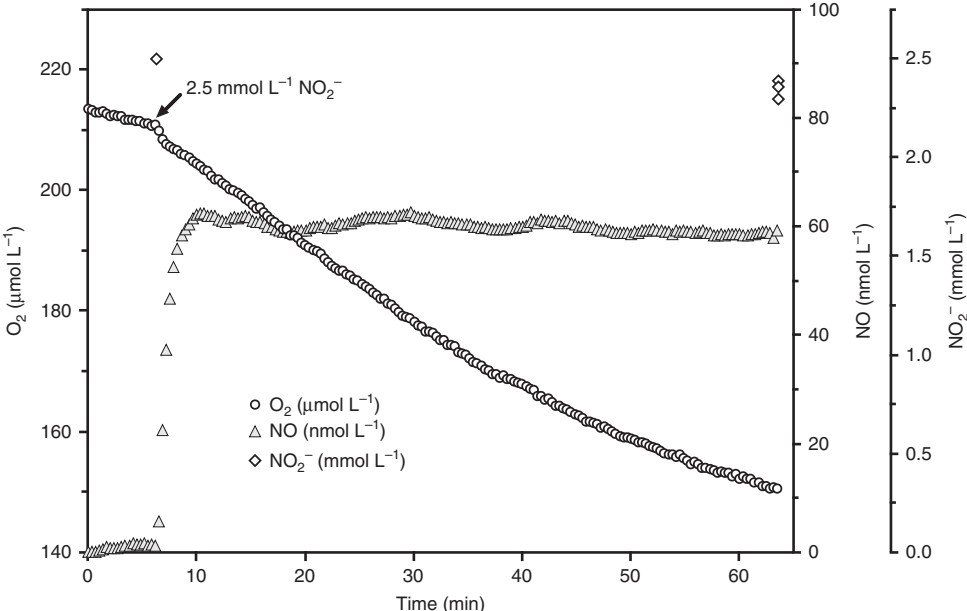

**Fig. 4** Instantaneous $O_2$ consumption and NO production during $NO_2^-$ oxidation by *N. inopinata*. The data shown here represents a single replicate of three biological replicates ($n = 3$). Two additional biological replicates are shown in Supplementary Fig. 8. Dissolved $O_2$ is shown in open circles, dissolved NO in filled gray triangles, and $NO_2^-$ in open diamonds. The $NO_2^-$ concentration immediately after injection (~6 min) were inferred from the injected volume of a stock $NaNO_2$ solution, otherwise $NO_2^-$ concentrations were determined in three technical replicates ($n = 3$). Experiments were performed in a 10-mL microrespiration (MR) chamber fitted with an $O_2$ and NO microsensors. The arrow marks the addition of 2.5 mM $NO_2^-$ into the MR chamber. No NO formation from $NO_2^-$ was measurable in oxic sterile media controls containing heat-killed biomass. Source data are provided as a Source Data file

effective maximal concentrations ($EC_{50}$) of 18.9 and 63.6 μM, respectively. The $EC_{50}$ of PTIO for *N. maritimus* and *N. viennensis* ranges from 17.5–18.3 μM, while *N. multiformis* and other AOB are only inhibited by PTIO concentrations >300 μM[51,56]. Washing and reharvesting *N. inopinata* cells treated with low but inhibitory concentrations of PTIO (33–100 μM PTIO) to remove the PTIO resulted in complete recovery (>95%) of activity, while cells treated with high concentrations of PTIO only showed partial (mean ± standard deviation: 28.3 ± 11.9%, 330 μM PTIO) or no recovery of activity (1.8 ± 3.7%, 1000 μM PTIO; Supplementary Fig. 5). These results are consistent with PTIO acting as a NO-binding, reversible inhibitor at low concentrations, and a irreversible cytotoxin at high concentrations. However, it still remains a possibility that the PTIO-dependent inhibition we observed was caused by the imino nitroxides (PTIs) and $NO_2$ formed by the reaction of PTIO and NO rather than NO chelation.

To determine whether PTIO exhibits general cytotoxicity toward *Nitrospira* cells, we tested the effect of PTIO on instantaneous $O_2$ consumption and growth in the non-ammonia oxidizing *N. moscoviensis*. Generally, PTIO was significantly less inhibitory to *N. moscoviensis*, with estimated $EC_{50}$ values of 385.6 μM and 100.2 μM for instantaneous activity and batch culture $NO_2^-$ oxidation, respectively (Supplementary Fig. 5). However, considering that PTIO was not fully inhibitory to *N. moscoviensis* at even 1000 μM PTIO during batch culture growth, the calculated $EC_{50}$ in this condition is not reliable. Nearly complete inhibition (<10% activity) was only evident when we measured instantaneous $O_2$ consumption in the 1000 μM PTIO treatment and the calculated $EC_{50}$ under these conditions (385.6 μM) is very similar to that of previously published values for AOB[51]. Finally, washing and reharvesting *N. moscoviensis* cells that were significantly inhibited by PTIO resulted in no measurable recovery when compared to an untreated control (6.3 ± 3.3%, 1000 μM PTIO + wash; Supplementary Fig. 5). Collectively, the inhibitory effect of PTIO on *N.*

*moscoviensis* at these concentrations is permanent even after it is removed and this suggests that PTIO at high concentrations acts as a irreversible cytotoxin in *N. moscoviensis*.

Taken together, the NO production profile, the low $EC_{50}$ of PTIO, and the reversibility of PTIO toxicity in *N. inopinata* suggest an essential role for NO as an intermediate in *N. inopinata*. Furthermore, our data reveal that PTIO can no longer be considered as a selective inhibitor of AOA in studies targeted at assessing the roles of various nitrifying microbes in environmental systems as comammox *Nitrospira* will also be inhibited upon addition of low concentrations of this NO scavenger. Furthermore, inhibition of other nitrifying microbes not mediated by NO scavenging but by the potential cytotoxicity of PTIO, especially at higher concentrations, has to be considered in the design of such molecular ecology experiments.

**Respirometry suggests abiotic $N_2O$ formation by *N. inopinata*.** AOB and AOA release small amounts of $NH_2OH$ during ammonia oxidation under fully oxic conditions and the abiotic conversion of $NH_2OH$ with $Fe^{3+}$, $Mn^{4+}$, $Cu^{2+}$, $NO_2^-$, and other components of the surrounding matrix under oxic conditions[57] explains most of the $N_2O$ formation by these nitrifiers under these conditions[6,7,10]. Hypoxia leads to a significant increase in the $N_2O$ yield from $NH_3$ in AOB[9,12,13,58], while oxygen concentration has no major influence on the $N_2O$ yields of all tested AOA[8,17,52]. Main pathways that contribute to $N_2O$ formation during $O_2$ limitation in these nitrifiers are nitrifier-denitrification –the sequential enzymatic reduction of $NO_2^-$ to $N_2O$ via NIR and NOR – and chemodenitrification, whereby $NO_2^-$ or NO is non-enzymatically reduced to $N_2O$ via media components or heat-killed cell moieties[6,59]. The NO can be derived from enzymatic $NO_2^-$ reduction or from reactions of $NO_x$ with $NH_2OH$. Studies on AOB from diverse environments and adapted to different substrate concentrations showed that all of them produce $N_2O$ within minutes of the transition from oxic to anoxic

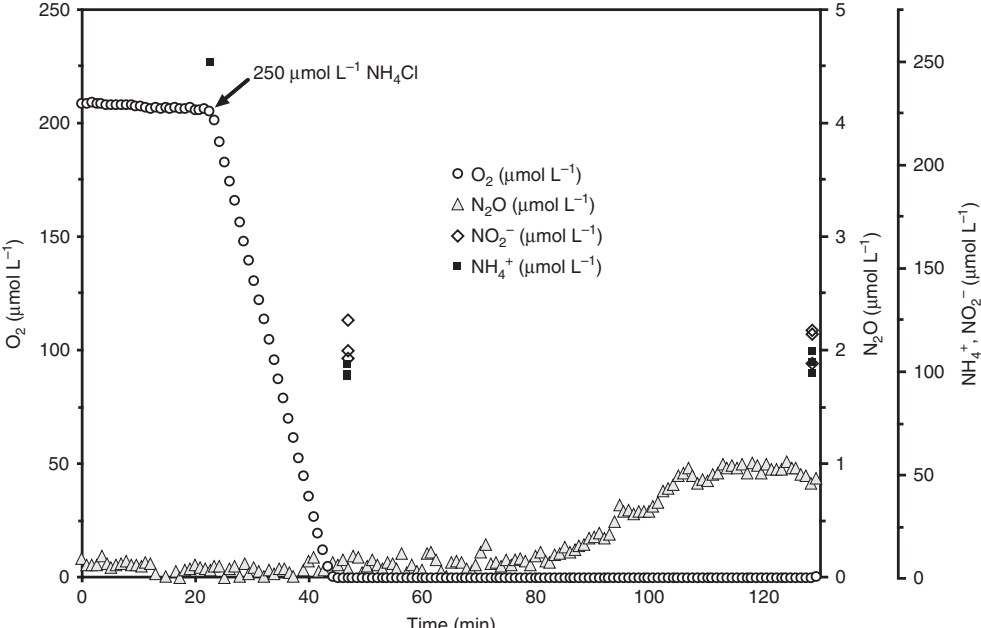

**Fig. 5** Instantaneous $O_2$ consumption and $N_2O$ production during $NH_3$ oxidation by *N. inopinata*. The data shown here is a single representative of three biological replicates ($n = 3$). Two additional biological replicates are shown in Supplementary Fig. 9. Dissolved $O_2$ is shown in open circles, dissolved $N_2O$ in filled gray triangles, $NO_2^-$ in open diamonds, and $NH_4^+$ in filled squares. The $NH_4^+$ concentration immediately after injection (~23 min) was inferred from the injected volume of a stock $NH_4Cl$ solution, otherwise $NO_2^-$ and $NH_4^+$ concentrations were determined in three technical replicates ($n = 3$). Experiments were performed in a 10-mL microrespiration (MR) chamber fitted with $O_2$ and $N_2O$ microsensors. The arrow marks the addition of 250 μM $NH_4Cl$ into the MR chamber. About 110 μM residual $NO_2^-$ was present in the chamber once $O_2$ reached below the detectable level at ~45 min. Source data are provided as a Source Data file

conditions[6]. This released $N_2O$ originates mainly from nitrifier-denitrification in AOB that encode nitric oxide reductases (*norCBQD* or *norSY*) and to a much lower extent from chemo-denitrification in oligotrophic AOB that lack NO reductases and emit large quantities of NO during transition from oxic to anoxic conditions[6]. The soil thaumarcheaon *N. viennensis*, which cannot perform nitrifier-denitrification to $N_2O$, also releases large quantities of NO during hypoxic conditions which then reacts abiotically with medium components ($Cu^{2+}$ and $Fe^{2+}$) to form $N_2O$[8,10].

Using $NH_3$ as the electron donor, we measured instantaneous $N_2O$ production with microsensors by *N. inopinata* biomass during oxic conditions and through a period of hypoxia. Interestingly, *N. inopinata* did not produce $N_2O$ during aerobic $NH_3$ oxidation or during the transition from oxic to anoxic conditions (Fig. 5) that could be measured with the $N_2O$ microsensor that has a relatively low sensitivity (100 nM). Small quantities of $N_2O$ were only measurable with the sensor from *N. inopinata* ~40 min after $O_2$ was depleted below the limit of detection (~300 nM dissolved $O_2$). This delayed $N_2O$ accumulation did not coincide with NO release in replicate traces where NO was measured (Figs. 3 and S6). Unlike growth assays, these short micro-respirometry experiments with high biomass and high substrate concentrations likely result in a relatively large accumulation in intracellular reductant in the form of $NH_2OH$[52]. To test whether this delayed $N_2O$ release could originate from cell lysis and abiotic formation of $N_2O$ from $NO_2^-$, $NH_2OH$, and other media components, we incubated heat-killed *N. inopinata* cells under anoxic conditions in AFW media supplemented with 1.8 μM $NH_2OH$ and 100 μM $NO_2^-$. This $NH_2OH$ concentration was chosen based on determined extracellular $NH_2OH$ concentrations in enrichment cultures of *N. inopinata*[7]. These abiotic controls yielded 0.83 ± 0.1 μM $N_2O$ (mean ± standard deviation), explaining most of the observed $N_2O$ in the live cell incubations.

Together, these results suggest that the delayed $N_2O$ formation under $O_2$-limiting conditions by *N. inopinata* originates mainly from abiotic reactions of accumulated $NH_2OH$ and not from the enzymatic reduction of NO to $N_2O$.

**N. inopinata $N_2O$ originates from abiotic $NH_2OH$ conversion.** The short duration of the micro-respirometry experiments and the limited sensitivity of the $N_2O$ microsensors (100 nM dissolved $N_2O$) makes calculating the precise $N_2O$ yield during $NH_3$ oxidation very difficult. To calculate the precise yields of $N_2O$ (as a $N_2O/NH_3$ ratio in percent) we performed batch growth experiments with *N. inopinata* biomass in sealed serum vials under $NH_3$- and $O_2$-limiting conditions and measured consumption of relevant metabolites and production of $N_2O$ using gas chromatography. $N_2O$ formation during oxic (~20.8% $O_2$), $NH_3$-limited growth was entirely dependent on $NH_3$ oxidation; no $N_2O$ formation was observed during $NO_2^-$ oxidation to $NO_3^-$ after all of the $NH_3$ was depleted (Fig. 6a). Similarly, $NO_2^-$ oxidation by *N. moscoviensis* in batch growth experiments yielded only trace (mean±standard deviation: 4.3 ± 2.3 nM) amounts of $N_2O$ under fully oxic and hypoxic conditions (Supplementary Fig. 7).

The $N_2O$ yield (as a $N_2O$ per $NH_3$ ratio %) during aerobic $NH_3$ oxidation for *N. inopinata* was 0.070 ± 0.006 (mean ± standard deviation). This measured $N_2O$ yield closely matched the predicted yield of $N_2O$ from abiotic reactions between media components and extracellular $NH_2OH$ for *N. inopinata* cultivated at 500 μM $NH_4^+$ (0.06%)[7]. This suggests that all of the $N_2O$ produced by *N. inopinata* during oxic growth on $NH_3$ came from the abiotic conversion of $NH_2OH$. Further, an $N_2O$ yield of 0.073 ± 0.033 % in $O_2^-$ limited incubations that were initiated at ~1.8 μM dissolved $O_2$ (equivalent to 0.89% $O_2$) demonstrated that the $N_2O$ yield was not influenced by hypoxia (Fig. 6b). This matches observations in *N. maritimus* and *N. viennensis* in which the $N_2O$ yield during growth

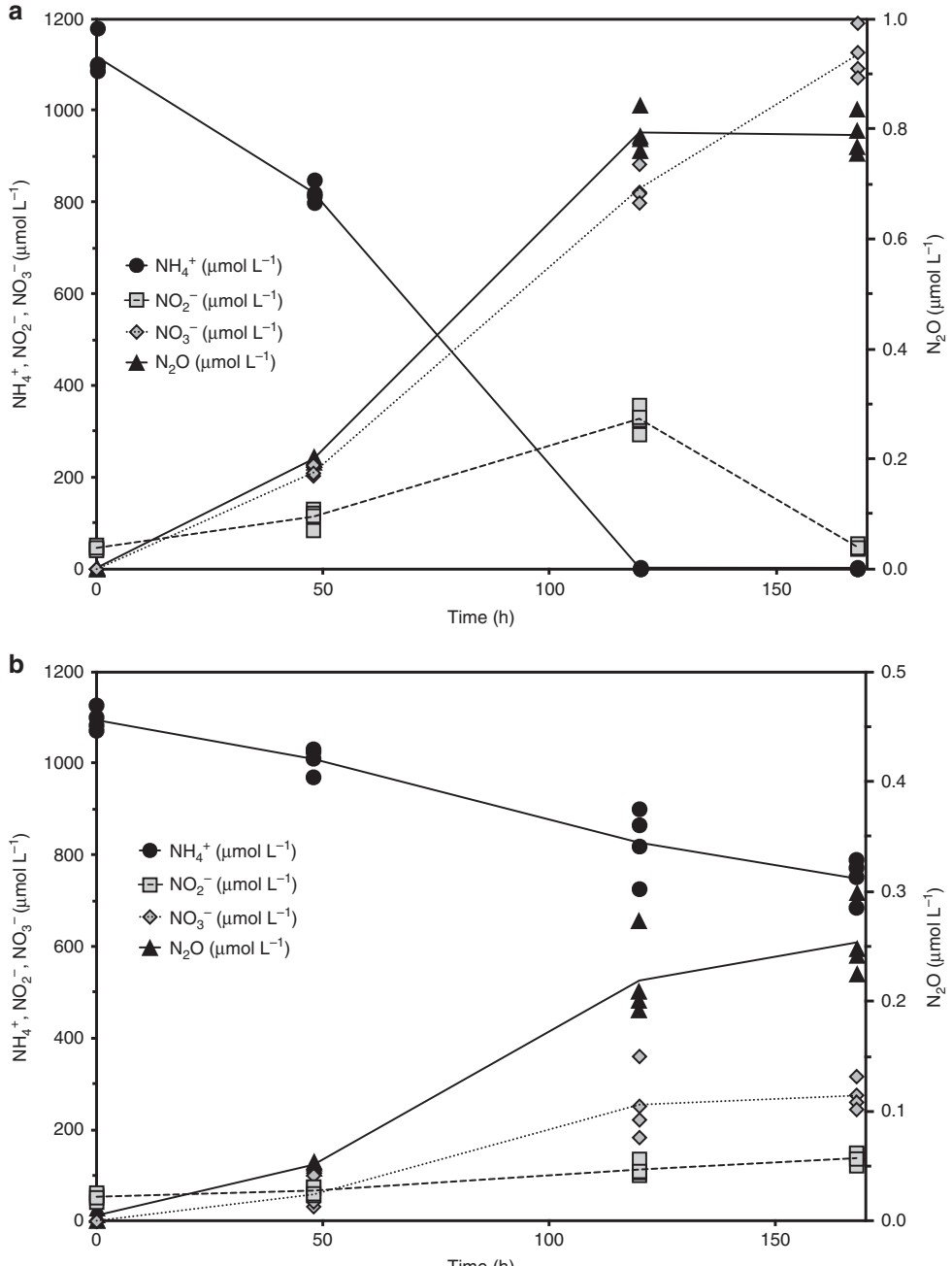

**Fig. 6** Oxidation of $NH_3$ to $NO_2^-$ and $NO_3^-$ and concurrent $N_2O$ yield during growth of *N. inopinata*. *N. inopinata* was incubated in closed serum vials under fully oxic conditions (initial $O_2$ at ~20.8%, **a**) or hypoxic conditions (initial $O_2$ at ~0.89%, **b**). $NH_4^+$ is shown in filled circles, $NO_2^-$ in filled squares, $NO_3^-$ in filled diamonds, and $N_2O$ in filled triangles; for each time point, individual replicates ($n = 4$ biological replicates) are plotted and the means connected by a solid black line. The amount of biomass at $t = 0$ was the same in the oxic and hypoxic vials. $NH_3$ oxidation to $NO_3^-$ is nearly stoichiometric in panel **a** due to unlimiting $O_2$; in panel **b**, however, limiting $O_2$ concentrations prevented the stoichiometric oxidation of $NH_3$ to $NO_3^-$. $O_2$ concentrations in the hypoxic treatment at 168 h were below the limit of detection (300 nM). Source data are provided as a Source Data file

is not influenced by varying $O_2$ concentrations and is in stark contrast to AOB like *N. europaea* and *N. multiformis* that produce ~3 times more $N_2O$ on a per mol $NH_3$ basis during nitrifier-denitrification under low $O_2$ conditions[8,9,13,58]. Furthermore, the $N_2O$ yield values for *N. inopinata* ($0.070 \pm 0.006$%) are very similar to those observed for *N. viennensis* (0.07 to 0.09%) and other several other soil AOA (~0.080%)[8].

To obtain additional information on the pathway(s) leading to $N_2O$ production in *N. inopinata*, we determined the intramolecular (natural abundance) distribution of $^{15}N$ within the linear

$N_2O$ molecule ($N^\beta$-$N^\alpha$-O) between the central ($\alpha$) and the external ($\beta$) position (also called the site preference SP; $\delta^{15}N$-SP, defined as $\delta^{15}N^\alpha$ - $\delta^{15}N^\beta$) from headspace $N_2O$ harvested from batch yield experiments performed under $NH_3$- and $O_2$-limiting conditions (Fig. 6). The site preference of $N_2O$ produced during $NH_3$-limited growth (mean ± standard deviation: $33.4 \pm 0.3$‰) was the same as during hypoxia ($34.2 \pm 1.4$‰). The unvarying SP of $N_2O$ produced during the different conditions reflects that the $N_2O$ source did not change in response to hypoxia. Such a positive $\delta^{15}N$-SP is also consistent with a $N_2O$ source from

$NH_2OH$ conversion (observed under oxic conditions to be 30.8–35.6‰[60,61], for AOB and 13.1–30.8‰ for AOA[16,62]), whereas $N_2O$ produced via nitrifier-denitrification or heterotrophic denitrification has a SP near or below 0[61,63,64].

In summary, we first show that *N. inopinata* has very tight control over NO production during aerobic $NH_3$ oxidation, that it also produces NO during aerobic $NO_2^-$ oxidation, but does not release NO during hypoxia. These data collectively strongly suggest an important but not yet fully explored metabolic role of NO for complete nitrifiers of the genus *Nitrospira*. Consistently, low concentrations of the NO scavenger PTIO inhibit *N. inopinata*. Importantly, we then demonstrate that *N. inopinata* cannot denitrify to $N_2O$ due to the absence of NO reductases in the genome, that it has a low $N_2O$ yield that is comparable to cultivated soil AOA and not influenced by $O_2$ levels, and that the $N_2O$ formed originates from abiotic $NH_2OH$ conversion. While we cannot rule out the possibility that some comammox *Nitrospira* encode a currently unknown NO reductase, we hypothesize that all other currently enriched and/or metagenomically characterized comammox *Nitrospira* also lack the ability to denitrify to $N_2O$ because they lack homologs for known NO reductases. Consequently, adoption of conditions that favor the growth of complete nitrifiers over AOB, for example, in engineered systems and in soils where comammox bacteria have been identified[28,30,65], may influence nitrification-dependent $N_2O$ emissions. Quantifying the relative contribution of comammox organisms as well as canonical nitrifiers to ammonia oxidation and $N_2O$ production in natural and engineered environments will thus be an important aim for follow-up research.

## Methods

**Strains and cultivation**. *Nitrospira inopinata* was routinely cultivated at 37 °C in AOM medium[24] which contained (per L): 50 mg $KH_2PO_4$, 75 mg KCl, 50 mg $MgSO_4 \cdot 7H_2O$, 584 mg NaCl, 1 mL specific trace element solution (TES), and 1 mL of selenium-tungstate solution (SWS). TES contained (per litre of deionized water): 34.4 mg $MnSO_4 \times H_2O$, 50 mg $H_3BO_3$, 70 mg $ZnCl_2$, 72.6 mg $Na_2MoO_4 \times 2H_2O$, 20 mg $CuCl_2 \times 2H_2O$, 24 mg $NiCl_2 \times 6H_2O$, 80 mg $CoCl_2 \times 6H_2O$, and 1 g of $FeSO_4 \times 7H_2O$ (dissolved in 2.5 mL 37% HCl)[24]. SWS contained (per litre): 0.5 NaOH, 3 mg $Na_2SeO_3 \times 5H_2O$, and 4 mg $Na_2WO_4 \times 2H_2O$[24]. The AOM medium was supplemented with 1 mM $NH_4^+$ and 4 g L$^{-1}$ of $CaCO_3$, the latter of which acts as a solid buffering system and substrate for attachment. The final pH of the medium was ~8.0 after sterilization. AOM medium could not be used to cultivate *N. inopinata* for the micro-respirometry experiments because the large quantities of undissolved $CaCO_3$ interfered mechanically with the highly sensitive NO sensor that, unlike the $O_2$ and $N_2O$ sensors, has an extremely small signal ($10^{-14}$ ampere). Consequently, Artificial Fresh Water Medium (AFW) was used to prepare *N. inopinata* biomass for the micro-respirometry (MR) experiments. The basal AFW (without pyruvate) containing 1 mM $NH_4^+$ was supplemented with (per L): 1 mL of 1000X non-chelated trace element solution, 1 mL of 1000X vitamin solution, and 1 mL of 7.5 mM Fe-NaEDTA solution[16]. The final pH of the AFW was set to ~7.2 through the addition of HEPES (prepared initially to pH 7.6) and $NaHCO_3$, which were added at final concentrations of 4 mM and 3 mM, respectively. The pure culture of *Nitrospira inopinata* has been deposited in the JCM (accession no. JCM 31988). *Nitrosomonas europaea* ATCC 19718 was cultivated in the same basal AFW medium at pH 8.5 and supplemented with 2.5 mM total $NH_4Cl$. *Nitrospira moscoviensis* was cultivated at 37 °C in mineral nitrite medium (NOM)[66] amended with 5 mM $NO_2^-$. Basal NOM contained (per litre of deionized water): 10 mg $CaCO_3$, 0.5 g NaCl, 50 mg of $MgSO_4 \times 7H_2O$, 0.15 g $KH_2PO_4$, 10 mg $NH_4Cl$, 0.034 mg $MnSO_4 \times H_2O$, 0.05 mg $H_3BO_3$, 0.07 mg $ZnCl_2$, 0.0726 mg $Na_2MoO_4 \times 2H_2O$, 0.02 mg $CuCl_2 \times 2H_2O$, 0.024 mg $NiCl_2 \times 6H_2O$, 0.08 mg $CoCl_2 \times 6H_2O$, and 1 mg $FeSO_4 \times 7H_2O$[66].

**Instantaneous NO and $N_2O$ measurement**. *N. inopinata*, *N. moscoviensis*, and *N. europaea* cultures were monitored daily and harvested immediately once all the substrate was consumed (normally ~7–9 days) by centrifugation using 10 kDa-cutoff, Ultra-15 Centrifuge Filter units (Amicon, Darmstadt, Germany). About 500 mL of mid-exponential phase culture was harvested per replicate experiment for *N. inopinata*, while 300 L of culture was used per replicate experiment with *N. moscoviensis* and *N. europaea*. Harvested biomass was washed twice with substrate-free medium and then resuspended in 10 mL of the same substrate-free medium. The biomass was then transferred into a 10 mL, double-port MR chamber (allowing no headspace) that was fitted with two MR injection lids and two glass coated stir bars. All MR experiments were performed in a recirculating water-bath at 37 °C. $O_2$ uptake was

measured using a OX-MR oxygen microsensor (Unisense, Aarhus, Denmark). $N_2O$ and NO concentrations were measured using an $N_2O$-MR sensor (Unisense) and an ami700-NO sensor (Innovative Instruments, Inc., Tampa, USA), respectively. Substrate ($NH_4^+$ or $NO_2^-$) were injected into the chamber via an injection port using either a 10-μL or 50-μL syringe (Hamilton, Reno, USA) fitted with a 26G needle. A 250-μL syringe (Hamilton, Reno, USA) fitted with a 26 G needle was also used to withdraw small aliquots (150 μL) from MR chambers during experiments to measure $NH_4^+$ and $NO_2^-$ concentrations, and sterile media was always backfilled. Starting metabolite concentrations were inferred from injected amounts; otherwise, all $NH_4^+$ and $NO_2^-$ concentrations were measured. $NH_4^+$, $NO_2^-$, and $NO_3^-$ concentrations were quantified photometrically with the Berthelot reagent, acidic Griess reagent, and $VCl_2$/Griess reagent, respectively[24,33], using an Infinite 200 Pro spectrophotometer (Tecan Group AG, Maennedorf, Switzerland). $NH_4^+$ injections into the 10 mL MR chamber were always 250 μM for *N. inopinata* and 2 mM for *N. europaea*. $NO_2^-$ injections were 1 mM for *N. moscoviensis* and 2.5 mM for *N. inopinata*. Higher concentrations of $NO_2^-$ were used for *N. inopinata* due to the comparatively low apparent affinity of *N. inopinata* for $NO_2^-$ ($K_{m(app)} = 449.2 \pm 65.8$ μM) compared to its very high affinity for ammonia ($K_{m(app)} = 63 \pm 10$ nM $NH_3$)[33]. Further, it was not feasible to measure $NO_2^-$ oxidation from fully oxic conditions to anoxic conditions due to the slow rate of $NO_2^-$-dependent $O_2$ uptake and the poor affinity of *N. inopinata* for $NO_2^-$. We expected no cell doubling during the MR experiments, as the doubling time of all strains (10–40 h) is significantly greater than the duration of the longest MR trace (~2 h). The OX-MR and $N_2O$-MR sensors were plugged directly into a microsensor multimeter while the ami700-NO sensor was polarized using a One-Channel Free Radical Analyzer (World Precision Instruments, Sarasota, USA). All electrodes were polarized for >1 day prior to use and calibrated according to the manufacturer's instructions. All data were logged on a laptop via the microsensor multimeter using SensorTrace Logger software (Unisense). The output from One-Channel Free Radical Analyzer was run into the microsensor multimeter using a BNC/lemo adapter. Abiotic controls were performed in triplicate as described above but with heat-killed cells (autoclaved at 121 °C for 20 min) resuspended in sterile anoxic medium. Anoxically prepared aliquots of $NH_4^+$, $NO_2^-$ and $NH_2OH$ were injected into the MR chamber through the injection port using a 10 μL syringe (Hamilton, Reno, USA). Anoxic AFW, $NH_4^+$, $NO_2^-$, and $NH_2OH$ were prepared by sparging the solutions with $N_2$ gas for 1 h prior to use.

**PTIO inhibition**. Batch and micro-respirometry inhibition experiments with the NO scavenger 2–phenyl-4,4,5,5,-tetramethylimidazoline-1-oxyl 3-oxide (PTIO; purity >98%; TCI, Germany) were performed using exponential-phase cultures of *N. inopinata* and *N. moscoviensis* grown in AOM medium or mineral nitrite medium (NOM)[66], respectively. For the batch experiments, cells were harvested by centrifugation ($8000 \times g$, 20 min, 20 °C), washed once (10 mg L$^{-1}$ $CaCO_3$ AOM medium for *N. inopinata* and $NO_2^-$-free NOM medium for *N. moscoviensis*) and resuspended undiluted in the same medium containing either 1 mM $NH_4Cl$ (*N. inopinata*) or 1 mM $NO_2^-$ (*N. moscoviensis*). The cultures was aliquoted (20 mL) in glass serum bottles (60 mL) sealed with crimp caps and quadruplicates were incubated in the dark at 37 °C in the presence of 0, 3.3, 10, 33, 100, and 330 μM PTIO, respectively. Only the concentrations of PTIO that inhibited *N. inopinata* were tested on *N. moscoviensis* (in addition to the unamended control) - 0, 33, 100, and 330 μM PTIO. PTIO was added in the respective amounts as a 10-mM stock solution in autoclaved MilliQ water. Abiotic controls with 0 and 330 μM PTIO and controls with heat-killed cells with 0 μM PTIO were run in triplicate. $NH_4^+$, $NO_2^-$, concentrations were quantified with the Berthelot reagent, acidic Griess reagent, and $VCl_2$/Griess reagent, respectively[24,33], using an Infinite 200 Pro spectrophotometer (Tecan Group AG, Maennedorf, Switzerland). Activity percentage was calculated by comparing the difference in rate of $NH_3$ (for *N. inopinata*) or $NO_2^-$ (for *N. moscoviensis*) oxidation during the linear section of the substrate oxidation curve between the non-inhibited control culture and the cultures exposed to the various concentrations of PTIO.

For the micro-respirometry-based PTIO inhibition experiments, 360 mL of exponential phase culture was harvested using 10 kDa-cutoff, Ultra–15 Centrifuge Filter units as described above, washed twice with N-free medium, homogenized by vigorous vortexing, and split into 25 2.5 mL aliquots. Individual aliquots were then amended with 0, 33, 100, 330, or 1000 μM PTIO and incubated at 37 °C for at least 2 h. For each experiment, an aliquot was transferred into a 2 mL, single-port MR chamber containing a glass-coated stir bar. $O_2$ uptake was measured using OX-MR oxygen microsensors in a recirculating water-bath at 37 °C and stirring at 400 RPM. $NO_2^-$ was injected into the chamber as described above. Viability of the harvested biomass over the length of the entire experiment (~8 h) was confirmed by measuring the rate of substrate-dependent $O_2$ uptake in biomass incubated at 37 °C without PTIO at the end of the experiment. To determine if removal of PTIO after PTIO treatment restored activity (as being expected if inhibition occurred due to NO scavenging), we harvested the biomass treated with inhibitory concentrations of PTIO using Ultra-15 Centrifuge Filter units, washed it twice with $NO_2^-$-free mineral medium using the same Ultra-15 Centrifuge filter units, resuspended it in $NO_2^-$-free mineral medium, and transferred it to a 2 mL MR chamber for re-measurement of substrate-dependent $O_2$ uptake. To control for activity loss due to biomass loss or additional centrifugation, biomass from the 0 μM PTIO treatment group was treated identically. All experiments were run in triplicate. Activity percentage was calculated by comparing the difference in rate of

$NH_3$ (for *N. inopinata*) or $NO_2^-$ (for *N. moscoviensis*) oxidation during the linear section of the substrate oxidation curve between the non-inhibited control culture and the cultures exposed to the various concentrations of PTIO.

**$N_2O$ yield**. Late exponential-phase cultures (containing ~0 μM $NH_4^+$ and ~350 μM remaining $NO_2^-$) of *N. inopinata* grown in FWM (described above) were transferred at 10% (final volume of 139 mL) to new FWM containing ~1.1 mM $NH_4^+$ and aliquoted into glass serum bottles (~255 mL) sealed with butyl rubber stoppers and crimp caps. Quadruplicates were incubated in the dark at 37 °C under a fully oxic atmosphere, with lab air as the headspace, and at low $O_2$ conditions (~0.9% $O_2$ with $N_2$ as the balance, equivalent to 1.8 μM dissolved $O_2$), respectively. A starting $O_2$ headspace concentration of ~0.9% in the hypoxic vials was chosen so the starting mol ratio of $NH_4^+$ to $O_2$ was 2.2:1, resulting in $O_2$-limiting growth conditions. Hypoxic conditions were achieved by sparging the sealed serum bottles with $O_2$-free $N_2$ gas for 1 h and then backfilling the headspace with 4.5 mL of lab air. The final concentration of $O_2$ in the headspace of the hypoxic vials at the beginning of the experiments was 0.89% as verified with a needle-piercing OX-NP $O_2$ microsensor (Unisense). In addition, abiotic controls containing 1.1 mM $NH_4^+$, 250 μM $NO_2^-$, and 1 μM $NH_2OH$ with and without heat-killed *N. inopinata* cells were run in parallel to control for abiotic formation of $N_2O$. Gas samples (15 mL) of the headspace were taken from the sealed serum bottles using a sterile syringe and transferred into sealed 12 mL exetainers for $N_2O$ and $O_2$ analysis. The serum bottles were backfilled with 15 mL of $O_2$-free $N_2$ gas after each sampling. $N_2O$ was quantified using a TRACE GC Ultra series gas chromatograph (ThermoFisher Scientific, Waltham, USA) equipped with a pulse discharge detector (ThermoFisher Scientific, Waltham, USA), a Porapak N column, and a Al/AS1310 autosampler (S +H Analytik GmbH, Moenchengladbach, Germany). Final calculated $N_2O$ concentrations take into account $N_2O$ loss due to sampling and are $N_2O$ emissions above experimentally determined background (atmospheric) $N_2O$ levels. $NH_4^+$, $NO_2^-$, and $NO_3^-$ concentrations were quantified as described previously.

For *N. moscoviensis*, late-exponential phase cultures were transferred at 10% to new NOM containing ~1.0 mM $NaNO_2$ and aliquoted into glass serum bottles sealed with butyl rubber stoppers and crimp caps. Triplicates were incubated in the dark at 37 °C under a fully oxic atmosphere, with lab air as the headspace, and at hypoxic conditions (~1.0% $O_2$). Gas samples of the headspace were taken at the beginning of the experiment and once all of the substrate was depleted, so no backfilling was necessary after sampling. $N_2O$ was quantified as described above. Final calculated $N_2O$ concentrations are $N_2O$ emissions above experimentally determined background (atmospheric) $N_2O$ levels.

**Analysis of $N_2O$ isotopic signatures**. For analysis of the isotopic composition of the $N_2O$ in the headspace of the *N. inopinata* vials, 4 mL of the headspace gas was transferred to 120 ml glass flasks filled with helium at 50 kPa overpressure. Before filling with helium, the flasks had been evacuated and flushed with helium four times. $N_2O$ was analyzed with an isotope ratio mass spectrometer (IRMS, IsoPrime 100, Elementar Analysensysteme, Hanau, Germany) coupled to an online pre-concentration unit (TraceGas, Elementar Analysensysteme) as described in detail in ref. [67]. Briefly, for pre-concentration, the $N_2O$ was transferred to a cold trap in liquid nitrogen with a transfer time of 20 min. Water, $CO_2$, and any potential CO in the sample gas were removed with magnesium perchlorate, carbosorb, and a CO oxidation catalyst (Sofnocat), respectively. Then the sample gas $N_2O$ was cryo-focused on a second cold trap, before the sample was remobilized again and separated isothermally at room temperature on a capillary column (PoraPLOT Q, 30 m length × 0.32 mm inner diameter, 10 μm coating, Varian). After GC separation, the $N_2O$ was introduced in a stream of helium to the IRMS via an open-split in continuous-flow mode. Mass-to-charge ratios ($m/z$) of $N_2O$ at 44 ($^{14}N^{14}N^{16}O$), 45 ($^{14}N^{15}N^{16}O$, $^{15}N^{14}N^{16}O$, and $^{14}N^{15}N^{17}O$), and 46 ($^{14}N^{14}N^{18}O$), as well as $NO^+$ fragment ions of $N_2O$ at $m/z$ 30 ($^{14}N^{16}O^+$) and 31 ($^{15}N^{16}O^+$) were measured simultaneously by the IRMS. The values of $\delta^{15}N^{bulk}$, $\delta^{15}N^\alpha$, and $\delta^{18}O$ were calculated from $m/z$ 44, 45, and 46, including a $^{17}O$ correction according to[68]. The $\delta^{15}N^\beta$ value was calculated as $\delta^{15}N^\beta = 2 \cdot \delta^{15}N^{bulk} - \delta^{15}N^\alpha$, and $^{15}N$ site preference (SP) as $SP = \delta^{15}N^\alpha - \delta^{15}N^\beta$. Calibration was performed as described in ref. [67]. Reproducibility was 0.2‰ for $\delta^{15}N^{bulk}$, 0.3‰ for $\delta^{18}O$, and 0.4‰ for $\delta^{15}N^\alpha$ and $\delta^{15}N^\beta$.

**Comparative genomics**. Genes encoding NirK, the NorCBQD complex, NorSY complex, CytL, CytS, and NcyA were initially identified by blastp and subsequently examined phylogenetically for annotation. Protein-coding genes from genomes of interest were screened using blastp against databases constructed from previously published lists[6]. For each gene set of interest, blastp results (default parameters) were filtered for queries that hit database entries with at least 70% of the query length. These putative genes were then aligned to the sequences that were used to construct the blastp databases using mafft[69] and examined for phylogenetic placement using FastTree2[70]. Neighborhoods of *nor* genes (*norCBQD* and *norSY*) were manually examined for synteny and scrutinized for genes missed by the blastp search. Alignments of CytL and CytS were further examined for the presence or absence of the diagnostic heme cross-linked lysine[49] to distinguish phylogenetic clades of CytL from CytS. Putative *nirK* genes were aligned to a set of previously published multicopper oxidase genes[53] and manually examined for phylogenetic placement into recognized clades of *nir*K.

**Reporting summary**. Further information on research design is available in the Nature Research Reporting Summary linked to this article.

## Data availability

The proteomics data used for Supplementary Fig. 3 is publicly available at the PRIDE/ProteomeXchange database under the accession code PXD013103 (ref. [24]). The source data underlying Figs. 3–6 and Supplementary Figs. 1, 2, 4–9 are provided as a Source Data file.

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

## Acknowledgements

We would like to thank Marton Palatinszky for providing *N. inopinata* biomass for the PTIO batch experiments. This work was funded by the Austrian Science Fund Grant P30570-B29 (to H.D. and M.W.), the ERC Advanced Grant NITRICARE 294343 (to M. W.), the Comammox Research Platform of the University of Vienna, and a Discovery Grant from the Natural Sciences and Engineering Research Council of Canada (to L.Y.S.).

## Author contributions

H.D. and M.W. conceived the study. H.D., M.W., and K.D.K. wrote the manuscript with input from all authors. K.D.K., M.Y.J., and C.J.S. performed the micro-respirometry and batch incubations. J.V. and P.P. performed the PTIO batch experiments. S.L. performed the batch *N. moscoviensis* experiments. C.H. made the comparative genome analysis. L.Y. S. assisted with data analysis. A.R., H.W., and N.B. performed the GC and $N_2O$ site preference measurements.

## Additional information

**Competing interests:** The authors declare no competing interests.

