## [Peer Review File · Nature Communications]

Reviewers' comments:

Reviewer #1 (Remarks to the Author):

Understanding the microbial processes, and the underlying metabolic pathways that govern the nitrogen cycling in the environment is critical for constraining the human impact on the nitrogen cycle. Furthermore, it is very interesting from physiological and biochemical aspects of microbial N transformation. The current manuscript addresses the physiological properties of one of the newest additions to the nitrogen cycle: the organisms that fully oxidize ammonium to nitrate in a single cell. The authors probe the pure culture of *N. inopinata* to gain insight into the ammonium and nitrite conversion pathways of this species. They detect that these cells produce and consume NO under ammonium oxidizing conditions, but do not produce N₂O to the extent of other aerobic ammonia-oxidizing bacteria. Under nitrite oxidizing conditions however, the cells exhibit net NO production. The authors also use omics approaches to speculate on which enzymes could be responsible for the measured activities.

The manuscript is well written, though at times it is repetitive (e.g. the parts where chemical production of N₂O is described) and too long to deliver the underlying message, which is that comammox process produces NO.

The authors state that NO production by these species is unexpected, however, their data on genomic insights and comparative analysis of the hydroxylamine dehydrogenase enzymes as well as earlier studies suggest that NO production is expected. The authors report that N₂O is produced to a lesser extent than "canonical" AOBs. The view of the authors is that it is most likely the measured N₂O is chemically produced. I find it hard to believe that a species making NO, which is a very toxic compound, leaves it to chemistry rather than keeping a tight control over its NO turnover by using dedicated enzymes. The major question that remains is, what would happen if NO was added directly to these cells? Would the cells be inhibited, would they produce more N₂O (maybe even enzymatically), or would they produce more nitrite? Another important question is whether these comammox species have a better coupling of their NO production and oxidation machinery than the canonical AOBs. This question can only be answered by direct purification of the involved enzymes. One last point concerning NO production under ammonia-oxidizing conditions is that the authors use literature data to compare their results with instead of performing the same experiments with pure cultures of canonical AOBs. I think it is usually fine to compare results with literature, but in this case, it is very important to see what the authors can measure when they subject canonical AOBs to the same experiments.

The results under-nitrite oxidizing conditions reveal that these species also produce NO when oxidizing nitrite as well. It also seems like they either do not consume this NO, or constantly turn NO over, but keep at an appreciable concentration of (60 nmol/L). This is indeed an odd observation. Why would cells produce a toxic compound and keep it around? Why would a species reduce nitrite to NO, and then have to re-oxidize it to nitrate via nitrite using two oxidation steps? Why would the same species that was so good in converting NO under ammonia-oxidizing conditions, now would not do the same here, but rather keep a constant NO level in the experimental setup? What about N₂O production under nitrite oxidizing conditions? Since the NO production phenomenon has not been observed under these conditions in other NOB, I think the authors should have extended their studies to known canonical NOBs.

The authors in the end state that growth conditions that would favor comammox over normal AOBs would benefit agroindustries or other engineered systems, and would/could result in less N₂O production. While this might be true, to the best of my knowledge, the conditions which are applied in agriculture or in wastewater treatment with respect to N-fluxes and the conditions which comammox species favor, are completely opposite. So it is difficult to envisage how conditions that could favor comammox could be created in agriculture or engineered ecosystems such as wastewater treatment plants. Finally, the authors state that the comammox species are widespread. Indeed, these species have been detected in several ecosystem, but their contribution to global N-cycling is not yet determined.

Reviewer #2 (Remarks to the Author):

The authors present an in-depth study of the ability of *Nitrospira inopinata*, a known comammox strain, to produce NO and N₂O (nitrous oxide) under controlled laboratory conditions. The work is put into context through a review of the pathways known to produce these gases in other nitrifying organisms. The work is detailed and adds important information to our ability to discover the sources of these important trace gases from both soil and wastewater systems. While in the article itself the author's conclusions are appropriately justified I found the supposition in the abstract that therefore comammox *Nitrospira* are only "minor contributors" to N₂O emissions too broad and not fully justified. The authors contend that these organisms are widespread and active in the environment. This means that they are assumed to play an important role in the production of nitrate. Since nitrate may be denitrified to N₂O then comammox logically may play an important role leading to nitrous oxide production. Although they are not directly producing much N₂O they do produce a key intermediate that may lead to nitrous oxide production. The section in question is the lines 37-40 in the abstract. I agree however that it would be worth testing whether encouraging the growth of comammox versus certain AOB in agricultural or engineered systems would result in the attenuation of nitrous oxide emissions. This is a minor point and the actual work and its presentation are a strong contribution to the ecophysiology of a novel group of environmentally relevant microbes.

We thank the reviewers for their careful reading of the manuscript, positive comments, constructive criticisms, and for helping us to greatly improve the revised paper which we now resubmit for review. Please find below our responses to the remarks of the reviewers as inserts (in bold):

Referee #1 (Remarks to the Author):

Understanding the microbial processes, and the underlying metabolic pathways that govern the nitrogen cycling in the environment is critical for constraining the human impact on the nitrogen cycle. Furthermore, it is very interesting from physiological and biochemical aspects of microbial N transformation. The current manuscript addresses the physiological properties of one of the newest additions to the nitrogen cycle: the organisms that fully oxidize ammonium to nitrate in a single cell. The authors probe the pure culture of *N. inopinata* to gain insight into the ammonium and nitrite conversion pathways of this species. They detect that these cells produce and consume NO under ammonium oxidizing conditions, but do not produce N₂O to the extent of other aerobic ammonia-oxidizing bacteria. Under nitrite oxidizing conditions however, the cells exhibit net NO production. The authors also use omics approaches to speculate on which enzymes could be responsible for the measured activities.

The manuscript is well written, though at times it is repetitive (e.g. the parts where chemical production of N₂O is described) and too long to deliver the underlying message, which is that comammox process produces NO. The authors state that NO production by these species is unexpected, however, their data on genomic insights and comparative analysis of the hydroxylamine dehydrogenase enzymes as well as earlier studies suggest that NO production is expected.

We thank the reviewer for this point. We completely agree that NO production by *N. inopinata* is expected from the HAO enzyme during ammonia oxidation. Therefore, we have changed the title of the manuscript by removing the dependence on NO. In addition, we have clarified the statement in the abstract that NO production from *N. inopinata* was unexpected (line numbers 39-40). The text (line numbers 137-146) also clarify that we expected NO production during ammonia oxidation from HAO. However, we found it highly unexpected that the *N. inopinata* NO production profile was much more similar to AOA than to AOB although the enzymatic repertoire of comammox is more closely related to AOB. Regarding NO, the unexpected findings were that *N. inopinata* i) produced less NO than AOB, ii) did not produce NO under hypoxic conditions (unlike AOA and AOB), and iii) was sensitive to inhibition by low concentrations of the NO-chelator PTIO (unlike AOB). Please note that the unique NO turnover pattern of *N. inopinata* even differs from those canonical AOB that also lack NO reductase (line numbers 198-202).

The authors report that N₂O is produced to a lesser extent than “canonical” AOBs. The view of the authors is that it is most likely the measured N₂O is chemically produced. I find it hard to believe that a species making NO, which is a very toxic compound, leaves it to chemistry rather than keeping a tight control over its NO turnover by using dedicated enzymes. The major question that remains is, what would happen if NO was added directly to these cells? Would the cells be inhibited, would they produce more N₂O (maybe even enzymatically), or would they produce more nitrite?

We agree that NO is very toxic and that its turnover has to be tightly controlled. Please note in this context that we did not conclude that NO would be chemically (non-enzymatically) degraded. We also do not directly link the chemical formation of N₂O (from hydroxylamine) to the turnover of NO. Such a link would not be supported by our data, which clearly show that the consumption of NO and formation of N₂O are asynchronous in *N. inopinata* (Figures 3 & 5 and Supplementary

Figures 6 & 9). In contrast, we observed (Fig. 3, Supplementary Figure 6) that *N. inopinata* produces less NO than oligotrophic and eutrophic AOB (line numbers 195-207 and Supplementary Figure 2), which demonstrates that *N. inopinata* has tighter control over NO turnover. This could very well be mediated by dedicated enzymes such as an unknown NO oxidoreductase (lines 143-146). Also, the measured NO emissions during ammonia and nitrite oxidation in *N. inopinata* are net emissions when the organism is challenged with a very high amount of substrate; 250 μM ammonium is higher than an oligotrophic organism like *N. inopinata* may normally experience. Based on the reviewer's suggestion, we have now quantified ammonia driven NO production in *N. inopinata* at low substrate concentrations (<15 μM ammonium). We show (Supplementary Figure 1) that lower and more realistic concentrations of substrate result in a largely reduced emission of NO during ammonia oxidation (< 0.8 nmol/L NO) (line numbers 189-192).

Another important question is whether these comammox species have a better coupling of their NO production and oxidation machinery than the canonical AOBs. This question can only be answered by direct purification of the involved enzymes. One last point concerning NO production under ammonia-oxidizing conditions is that the authors use literature data to compare their results with instead of performing the same experiments with pure cultures of canonical AOBs. I think it is usually fine to compare results with literature, but in this case, it is very important to see what the authors can measure when they subject canonical AOBs to the same experiments.

We thank the reviewer for the helpful suggestion. In the revised manuscript, we quantify NO production from *Nitrosomonas europaea* ATCC 19718 during ammonia oxidation (Supplementary Figure 2) and compare the results with *N. inopinata* (line numbers 202-207).

The results under-nitrite oxidizing conditions reveal that these species also produce NO when oxidizing nitrite as well. It also seems like they either do not consume this NO, or constantly turn NO over, but keep at an appreciable concentration of (60 nmol/L). This is indeed an odd observation. Why would cells produce a toxic compound and keep it around? Why would a species reduce nitrite to NO, and then have to re-oxidize it to nitrate via nitrite using two oxidation steps? Why would the same species that was so good in converting NO under ammonia-oxidizing conditions, now would not do the same here, but rather keep a constant NO level in the experimental setup?

We appreciate the valuable comment from the reviewer. While we do not yet fully understand the metabolic role of NO in comammox *Nitrospira*, NO production in nitrite oxidizing bacteria is not unprecedented. The nitrite oxidizing proteobacterium *Nitrobacter winogradskyi* is known to produce NO at appreciable concentrations (~ 65 nM) during growth on nitrite in a quorum sensing dependent manner (DOI: 10.1128/mBio.01753-16). *N. winogradskyi* is also a well described acyl-homoserine lactone (AHL) producer (DOI: 10.1128/AEM.01103-15). Several LuxI genes are present in the genome of *N. inopinata* and it is conceivable that the NO plays a role in biofilm formation or quorum sensing, as is well described for other biofilm forming bacteria (DOI: 10.1128/JB.00779-06), but we feel that this is too speculative to include in the current manuscript.

Stimulated by the comments of the reviewer, we have additionally measured NO and N₂O production from a canonical nitrite-oxidizer - *N. moscoviensis*. Though *N. moscoviensis* is a close relative of *N. inopinata*, NO emissions by *N. moscoviensis* during nitrite oxidation are below our limit of detection (0.25 nmol/L). Both *Nitrospira* encode *nirK* and lack any nitric oxide reductase homologs. In the context of the *N. moscoviensis* NO data, we have come up with an alternate hypothesis for the origin of NO in *N. inopinata* during nitrite oxidation; the high nitrite levels we used (which are required due to the low apparent affinity of the NXR for nitrite) may have

kinetically forced the comammox NO oxidoreductase enzyme (that is not required in canonical NOB) to function in the reverse direction - the production of NO rather than the oxidation of NO (produced by HAO) to nitrite. This could explain the differences in NO production during oxidation in both *Nitrospira*. We have added a discussion about this to the text to clarify the point brought up by the reviewer (line numbers 242-245).

What about N₂O production under nitrite oxidizing conditions?

In our N₂O yield experiments with *N. inopinata* (Fig. 6), there was no measurable N₂O production during nitrite oxidation to nitrate in the absence of ammonium (from 120 h to 168 h). It was not possible to do a growth yield experiment using nitrite as the sole electron donor because *N. inopinata* is unable to assimilate nitrite and therefore cannot grow on nitrite in the absence of ammonia (DOI: doi:10.1038/nature16461). Only trace quantities of N₂O (4.3 nM, equivalent to ~0.1 ppm) were formed in cultures of *N. moscoviensis* supplemented with 1 mM nitrite (Supplementary Figure 7) (line numbers 337-341).

Since the NO production phenomenon has not been observed under these conditions in other NOB, I think the authors should have extended their studies to known canonical NOBs.

We thank the reviewer for this comment and have now expanded the study (Supplementary Figures 4, 5, and 7) to include a canonical nitrite-oxidizing, non-comammox *Nitrospira* - *Nitrospira moscoviensis*. We measured NO production and N₂O yield during nitrite oxidation (Supplementary Figures 4 and 7) by *N. moscoviensis*. This closes a significant knowledge gap in the field, as NO and N₂O flux has never been measured from *Nitrospira*. The extremely low N₂O yield and total N₂O emissions from *N. moscoviensis* during nitrite oxidation are consistent with the lack of detectable N₂O production during nitrite oxidation in *N. inopinata* (line numbers 337-341). However, NO emissions by *N. moscoviensis* during nitrite oxidation are below our limit of detection (0.25 nmol/L, Supplementary Figure 4), which is in contrast to previous studies on *N. winogradskyi* and our study on *N. inopinata* (line numbers 224-232).

The authors in the end state that growth conditions that would favor comammox over normal AOBs would benefit agroindustries or other engineered systems, and would/could result in less N₂O production. While this might be true, to the best of my knowledge, the conditions which are applied in agriculture or in wastewater treatment with respect to N-fluxes and the conditions which comammox species favor, are completely opposite. So it is difficult to envisage how conditions that could favor comammox could be created in agriculture or engineered ecosystems such as wastewater treatment plants. Finally, the authors state that the comammox species are widespread. Indeed, these species have been detected in several ecosystem, but their contribution to global N-cycling is not yet determined.

Though comammox *Nitrospira* are well adapted to an oligotrophic lifestyle, they are highly abundant (and often dominate) in a large variety of wastewater treatment plants (WWTPs), soils, and freshwater systems. Recent studies have demonstrated clearly that comammox *Nitrospira* are the dominant ammonia oxidizers in various soils, sediments, tap water, coastal water, lake water, and leaf surfaces (DOI: 10.1128/AEM.01390-18). Critically, recent work has shown that comammox *Nitrospira* are sometimes more abundant than AOB and AOA in diverse wastewater treatment systems and activated sludge (DOI: 10.1016/j.biortech.2018.09.089). Additionally, it is also known that soil comammox *Nitrospira* are stimulated by increased long-term nitrogen deposition (doi: 10.1016/j.soilbio.2018.09.004.). The factors that select for comammox *Nitrospira* in these environments are not yet understood and further studies are urgently needed.

Taken together, comammox organisms can clearly be stimulated by factors other than oligotrophic conditions and utilizing these factors in engineered systems (i.e. selecting for biofilm formers) can disproportionately favor the growth of comammox organisms in these systems and reduce N₂O emissions.

Referee #2 (Remarks to the Author):

The authors present an in-depth study of the ability of *Nitrospira inopinata*, a known comammox strain, to produce NO and N₂O (nitrous oxide) under controlled laboratory conditions. The work is put into context through a review of the pathways known to produce these gases in other nitrifying organisms. The work is detailed and adds important information to our ability to discover the sources of these important trace gases from both soil and wastewater systems.

We thank the reviewer for these very positive comments on the general significance of our manuscript.

While in the article itself the author's conclusions are appropriately justified I found the supposition in the abstract that therefore comammox *Nitrospira* are only "minor contributors" to N₂O emissions too broad and not fully justified. The authors contend that these organisms are widespread and active in the environment. This means that they are assumed to play an important role in the production of nitrate. Since nitrate may be denitrified to N₂O then comammox logically may play an important role leading to nitrous oxide production.

We agree that comammox *Nitrospira* potentially play an important role in the production of nitrate (which may be denitrified by other microbes to N₂O). We have clarified the text (line number 42-43) to point out that we hypothesize that comammox *Nitrospira* produce less NO and N₂O than AOB only during the nitrification process.

Although they are not directly producing much N₂O they do produce a key intermediate that may lead to nitrous oxide production. The section in question is the lines 37-40 in the abstract. I agree however that it would be worth testing whether encouraging the growth of comammox versus certain AOB in agricultural or engineered systems would result in the attenuation of nitrous oxide emissions.

We agree that the NO produced by comammox *Nitrospira* could be converted abiotically or biotically to N₂O by other organisms in natural and engineered systems. However, the amounts of NO produced by comammox *Nitrospira* in our study were significantly lower than amounts of NO produced by AOB here (new Fig. S6) and reported in the literature. We have added text to clarify this point (line numbers 42-43).

This is a minor point and the actual work and its presentation are a strong contribution to the ecophysiology of a novel group of environmentally relevant microbes.

We thank the reviewer for these positive comments on the contribution that this work makes to the field.

REVIEWERS' COMMENTS:

Reviewer #1 (Remarks to the Author):

The manuscript has much improved. I appreciate the efforts of the authors. There are several points that still need to be addressed.

Lines 73-75: Please mention that the kinetics of CytL activity renders this protein irrelevant under physiological conditions.

Line 80: Energy conservation

Line 92: Protonated form, not acidic form

Line 172: Energy conservation

Line 180: This actually requires protein purification and characterization, not only physiological experiments. Please include this in this sentence.

Line 185-186: What exactly do the authors mean by instantaneous O₂ and NO kinetics? Do they mean kinetics of the reduction of these compounds upon their addition?

Line 215: Consumes instead of "re-consumes"

Line 228: Please call this nitrite oxidation to nitrate, not nitratation. Both terms are correct, but it would be better not to use nitratation to be consistent with the rest of the paper.

Line 234: Do the authors mean energy conservation through nitrite reduction? But copper-containing nitrite reductase (CuNIR, encoded by NirK) does not directly contribute to energy conservation in any organism. Please rephrase.

Line 265: "reversible" should precede "inhibitor", not "NO-binding"

Lines 267-274: This is a nice control experiment! However, *N. moscoviensis* does not produce considerable amounts of NO. Therefore, it could still be the case that PTIO inhibits *N. inopinata* due to the formation of nitrosylated compounds via the reaction of PTIO with NO instead of direct NO-scavenging action. Please include this possibility to the manuscript.

Line 285: PTIO should never have been used for this purpose. I am very happy to see that the authors show this.

Lines 311-313: The N₂O microsensors are sometimes also reactive towards NO. Did the authors consider this in their experiments?

Lines 374-376: Please include the possibility that there could be currently unknown NO-reducing enzymes.

The paper the authors mention in their rebuttal letter (DOI: 10.1016/j.biortech.2018.09.089) only show the presence of amoA genes of different AMO-encoding microorganisms in several wwtp. Therefore, it is not clear whether they actually actively contribute to ammonia oxidation in these engineered systems. It could be that they are there, but do not actively contribute to ammonia oxidation, it could be that they only oxidize nitrite in these systems. It could also be that not all comammox species behave the same way. Maybe some comammox species produce much higher N₂O, via currently unknown pathways, compared to AOB or AOA. For example, until recently, based on one AOA species it was assumed that all AOA have extremely high affinities to ammonia. However, in a very recent paper, written by many of the same authors as this manuscript, it was elegantly shown that this was not the case. Therefore, also considering that comammox species do not consume N₂O, I still do not agree with the authors that promoting comammox growth might bring down N₂O emissions. We do not have enough information to reach this conclusion. Therefore, please do not make this one of the main conclusions of the manuscript.

We are grateful to the referee for the positive comments, careful reading of the manuscript, and for constructive feedback that resulted in a greatly improved manuscript. Please find below our responses as inserts (in bold):

Reviewer #1 (Remarks to the Author):

The manuscript has much improved. I appreciate the efforts of the authors. There are several points that still need to be addressed.

Lines 73-75: Please mention that the kinetics of CytL activity renders this protein irrelevant under physiological conditions.

The change has been made as suggested (lines 100-101).

Line 80: Energy conservation

Thank you for this correction. The change has been made as suggested (line 106).

Line 92: Protonated form, not acidic form

Change made as suggested (line 119).

Line 172: Energy conservation

Change made as suggested (line 202).

Line 180: This actually requires protein purification and characterization, not only physiological experiments. Please include this in this sentence.

We completely agree that protein purification and characterization would have to be done in addition to whole cell physiological experiments and have amended the text accordingly (lines 211-212).

Line 185-186: What exactly do the authors mean by instantaneous O₂ and NO kinetics? Do they mean kinetics of the reduction of these compounds upon their addition?

In the context of the microsensors we used, instantaneous refers to continuous, sensitive, and high-resolution (i.e. in time) measurements of O₂, NO, and N₂O concentrations. Since these sensors allow us to measure small changes in concentration over short periods of time, the measurements are referred to as instantaneous.

Line 215: Consumes instead of “re-consumes”

Change made as suggested (line 247).

Line 228: Please call this nitrite oxidation to nitrate, not nitrataion. Both terms are correct, but it would be better not to use nitrataion to be consistent with the rest of the paper.

We have replaced “nitrataion” with “nitrite oxidation to nitrate” (lines 261-262).

Line 234: Do the authors mean energy conservation through nitrite reduction? But copper-containing nitrite reductase (CuNIR, encoded by NirK) does not directly contribute to energy conservation in any organism. Please rephrase.

We have rephrased the sentence as suggested to state that the NirK protein is postulated (within reference #36) to maintain redox balance by regulating electron flow, rather than being part of a complete denitrification pathway (lines 267-268).

Line 265: “reversible” should precede “inhibitor”, not “NO-binding”

We appreciate this correction. We have made the change as suggested (line 304).

Lines 267-274: This is a nice control experiment! However, *N. moscoviensis* does not produce considerable amounts of NO. Therefore, it could still be the case that PTIO inhibits *N. inopinata* due to the formation of nitrosylated compounds via the reaction of PTIO with NO instead of direct NO-scavenging action. Please include this possibility to the manuscript.

We agree that this possibility still exists and have modified the manuscript to include this (lines 305-307).

Line 285: PTIO should never have been used for this purpose. I am very happy to see that the authors show this.

We thank the reviewer for the positive comment.

Lines 311-313: The N₂O microsensors are sometimes also reactive towards NO. Did the authors consider this in their experiments?

We did consider this in our experimental design. While NO does have a small cross-reactivity with the N₂O sensor (~10%), the N₂O sensor is much less sensitive overall (100 nM) than the NO sensor (~0.25 nM). This means we would have to observe 1000 nM NO to get even a small response with the N₂O sensor and NO concentrations never reached > 60 nM in any experiment. Much more importantly, we measured NO and N₂O independently in replicate experiments under the same conditions and cross-reactivity of NO with the N₂O sensor cannot be an alternative explanation for our observations. For example, small quantities of N₂O were observed ~40 minutes after the onset of hypoxia in experiments with *N. inopinata* (using ammonium as electron donor) but this did not coincide with an NO production peak in the corresponding experiment where we measured NO. Thus, the measured N₂O could not be from interference from NO.

Lines 374-376: Please include the possibility that there could be currently unknown NO-reducing enzymes.

We completely agree that this is a possibility and have amended the manuscript to include this suggestion (lines 420-422).

The paper the authors mention in their rebuttal letter (DOI: 10.1016/j.biortech.2018.09.089) only show the presence of amoA genes of different AMO-encoding microorganisms in several wwtp. Therefore, it is not clear whether they actually actively contribute to ammonia oxidation in these engineered systems. It could be that they are there, but do not actively contribute to ammonia oxidation, it could be that they only oxidize nitrite in these systems. It could also be that not all comammox species behave the same way. Maybe some comammox species produce much higher N₂O, via currently unknown pathways, compared to AOB or AOA. For example, until recently, based on one AOA species it was assumed that all AOA have extremely high affinities to ammonia. However, in a very recent paper, written by many of the same authors as this manuscript, it was elegantly shown that this was not the case. Therefore, also considering that comammox species do not consume N₂O, I still do not agree with the authors that promoting comammox growth might bring down N₂O emissions. We do not have enough information to reach this conclusion. Therefore, please do not make this one of the main conclusions of the manuscript.

We thank the reviewer for the well-thought-out and constructive feedback. Based on the comment, we agree with the reviewer that currently unknown comammox species may have novel enzymes/pathways and that we cannot assert our previous conclusions based on the current data.

Therefore, we have significantly revised the last two sentences of the discussion; we have removed the previously mentioned conclusion that promoting comammox growth would bring down N₂O emissions and we instead suggest that the relative contribution of comammox *Nitrospira* to N₂O flux in engineered and natural systems is a topic that deserves investigation in the future (lines 426-435). Furthermore, we have revised the abstract to remove the conclusion that promoting growth of comammox organisms over AOB may help attenuate N₂O emissions (line 62).